# Early Performance of Recently Released Rootstocks with Grapefruit, Navel Orange, and Mandarin Scions under Endemic Huanglongbing Conditions in Florida

**J. Martin Zapien-Macias [1], Rhuanito Soranz Ferrarezi [2], Peter D. Spyke [3], William S. Castle [4], Frederick G. Gmitter, Jr. [4], Jude W. Grosser [4] and Lorenzo Rossi [1,\*]**

1 Indian River Research and Education Center, Horticultural Sciences Department, Institute of Food and Agricultural Sciences, University of Florida, Fort Pierce, FL 34945, USA

2 Department of Horticulture, University of Georgia, 1111 Plant Sciences Building, Athens, GA 30602, USA

3 Arapaho Citrus Management, Inc., Fort Pierce, FL 34945, USA

4 Citrus Research and Education Center, Horticultural Sciences Department, Institute of Food and Agricultural Sciences, University of Florida, Lake Alfred, FL 33850, USA

\* Correspondence: l.rossi@ufl.edu

**Abstract:** Huanglongbing (HLB), which is believed to be caused by the phloem-restricted bacterium *Candidatus* Liberibacter asiaticus (*C*Las), has decimated Florida's citrus production. Grapefruit production has declined 75%, mandarin 78%, and sweet orange 52% due to the high sensitivity of commercial scions and rootstocks to the disease. New combinations of scions and hybrid rootstocks may provide better performance than current commercial selections for Florida's fresh citrus production, particularly in the Indian River District. The objective of this study was to evaluate and compare University of Florida rootstocks and other recently released rootstocks grafted with grapefruit, navel orange, and mandarin scions by measuring tree growth and HLB tolerance. Three independent large-scale field trials were established in September 2019 in Fort Pierce, FL, USA. Trial 1 (T1) included 36 rootstocks with 'Ray Ruby' grapefruit as the scion; Trials 2 and 3 (T2 and T3, respectively) included 30 rootstocks with 'Glenn' navel orange F-56-11 and 'UF-950' mandarin as the scion, respectively. Tree canopy volume, trunk diameter, *C*Las titer, HLB severity index, and leaf nutrient concentrations were evaluated during 2020 and 2021. Significant differences among rootstock-scion combinations were found in each trial for most of the assessed traits. In T2, UFR-15 consistently developed the largest 'Ray Ruby' grapefruit trees during both years. In T3, 'Glenn' navel orange F-56-11 trees were larger on C-22, and US-802. Similarly, US-802 and US-942 generated the largest 'UF-950' mandarin trees. Overall, trees had optimum levels of macro- and micronutrients except for calcium. *C*Las infection and HLB visual index varied among scion-rootstock combinations, especially during the first year of growth when intensive flushing was produced. Generally, trees grew vigorously with WGFT+50-7 and Willits inducing the lowest HLB symptoms in all evaluations. Production and fruit quality need to be evaluated to determine the suitability of potential scion-rootstock combinations that can confer consistent economical and biological advantages under the current HLB scenario in the Indian River District.

**Keywords:** citrus greening; flatwoods; Indian River District

## 1. Introduction

Since HLB was reported in Florida in 2005, citrus production has dramatically downsized by approximately 67% due to the high disease sensitivity of commercial scions and rootstocks [1,2]. HLB is believed to be caused by the phloem-restricted bacterium *Candidatus* Liberibacter asiaticus (*C*Las) and vector-transmitted by the Asian citrus psyllid (*Diaphorina citri* Kuwayama) [3,4]. The disease affects more than 90% of Florida's citrus trees, resulting in rapid tree decline [5] and substantial economic losses [6–8]. To date, no

known cure exists for HLB. The Indian River District (IRD), a renowned grapefruit (*Citrus paradisi* Macf.) production area, accounts for nearly 80% of Florida's total grapefruit [2]. With the drastic HLB impacts on citrus production, citrus species diversification with navel oranges (*C. sinensis* [L.] Osb.) or mandarins (*C. reticulata* Blanco) for the fresh market and the use of superior rootstocks are potential strategies to minimize risks, strengthen profits, and secure the success of the modern citrus industry.

The Florida fresh-fruit citrus industry only represents 10% of the total citrus market in the state (excluding limes and lemons), with a calculated value of ~114 million dollars. The fresh citrus market is economically attractive as it represents roughly $4.4\times$ greater price than the processed fruit per box in Florida [9]. In Florida, 'Star Ruby', 'Ray Ruby' and 'Ruby Red' scions are the most popular propagated grapefruit cultivars due to their exceptional fruit quality attributes (red pigmented-fleshed, attractive pink blush in the rind, and sweet flavor) for the fresh market [10–12]. Navel and 'Valencia' oranges are popular as fresh fruit worldwide [13]. Recently, more than 46,000 navel trees were propagated in Florida, of which about 34% were 'Glenn' navel orange F-56-11 [12]. This nucellar selection and 'Cara Cara' red navel were the most propagated navel orange cultivars in Florida in 2020/21 [12]. Mandarins have gained attention as recent studies have suggested that a few commercial cultivars such as 'LB8-9' Sugar Belle® and 'Temple' showed HLB tolerance under natural HLB conditions in the IRD [14,15].

The use of rootstocks with superior horticultural (i.e., tolerance to biotic and abiotic factors) and economic (e.g., yield, fruit quality, and production window) attributes has played a key role in the evolution of citrus industries worldwide [16]. In citrus, HLB tolerance is defined as the ability of a *CLas*-infected tree to produce profitable quantities of fruit of acceptable quality [17].

Rootstocks have individual characteristics that contribute in positive or negative ways to the performance of a citrus tree [16,18]. Rootstock hybrids among *Poncirus trifoliata* (L.) Raf. and *Citrus* spp. have exhibited superior field performance under severe HLB conditions [19]. New breeding technologies such as somatic hybridization and molecular marker-assisted selection are broadening the possibilities for genetic manipulation, leading toward a new era of superior rootstocks that provide improved horticultural and economical attributes [16,20,21]. Some commercial materials, such as Kuharske citrange and x-639 are commercially propagated and used as parents in crosses to produce new rootstock candidates [22]. The effects of these and other HLB-tolerant rootstocks have not been fully explored, especially under flatwoods conditions (typically found in the IRD) and for fresh citrus market. In addition, the first years of tree development are critical to protecting citrus groves from HLB, as young trees flush more frequently which attracts the vector [3,4]. Thus, growing healthy trees during their early years of development can be crucial to securing the long-term productivity of a citrus grove. The objective of this study was to compare the early performance of trees consisting of University of Florida rootstocks (UFR), commercial and other recently generated rootstocks grafted with 'Ray Ruby' grapefruit, 'Glenn' navel orange F-56-11, and 'UF 950' mandarin by measuring tree growth and HLB tolerance.

## 2. Materials and Methods

### 2.1. Plant Material

Certified budwood and rootstock liners grown from seeds were provided by the Division of Plant Industry through the Florida Citrus Budwood Registration Bureau program. Trees of 'Ray Ruby' grapefruit, 'Glenn' navel orange F-56-11, and 'UF 950' mandarin were independently evaluated on UFR and recently released rootstocks that contained sexual and somatic hybrids from a wide range of different germplasms (Table 1). Trees were grown for approximately 8 months in 1-L pots with peat moss substrate media and maintained in a certified disease-free commercial nursery (Brite Leaf, Lake Panasoffkee, FL, USA).

**Table 1.** Rootstock cultivars employed in each field trial.

| Rootstock | Trial 1 | Trial 2 | Trial 3 | Origin | Parentage |
|---|:---:|:---:|:---:|:---:|---|
| UFR-1 | ● | ● | ● | UF | *C. reticulata* 'Nova' + *C. maxima* 'Hirado Buntan' × *C. reticulata* 'Cleopatra' + *P. trifoliata* 'Argentine' |
| UFR-2 | | ● | ● | UF | *C. reticulata* 'Nova' + *C. maxima* 'Hirado Buntan' × *C. reticulata* 'Cleopatra' + *P. trifoliata* 'Argentine' |
| UFR-4 | ● | ● | ● | UF | *C. reticulata* 'Nova' + *C. maxima* 'Hirado Buntan' × *C. reticulata* 'Cleopatra' + *P. trifoliata* 'Argentine' |
| UFR-5 | ● | ● | ● | UF | *C. reticulata* 'Nova' + *C. maxima* 'Hirado Buntan' × *C. sinensis* 'Succari' + *P. trifoliata* |
| UFR-15 | ● | ● | ● | UF | *C. maxima* 'Hirado Buntan' × *C. reticulata* 'Cleopatra' |
| UFR-16 | ● | ● | ● | UF | *C. maxima* 'Hirado Buntan' × *C. reticulata* 'Shekwasha' |
| UFR-17 | ● | ● | ● | UF | *C. reticulata* 'Nova' + *C. maxima* 'Hirado Buntan' × *C. aurantium* + 'Carrizo' citrange |
| WGFT+50-7 | ● | | ● | UF | *C. paradisi* 'White' + *P. trifoliata* '50-7' |
| 2247×2075-02-26 | ● | ● | ● | UF | *C. reticulata* 'Nova' + *C. maxima* 'Hirado Buntan' × *C. reticulata* 'Cleopatra' + 'Swingle' citrumelo |
| A+Volk×Orange 19-11-8 | ● | ● | ● | UF | *C. amblycarpa* + *C. volkameriana* × Orange 19 (*C. reticulata* 'Nova' + *C máxima* 'Hirado Buntan' × *P. trifoliata* 'Argentine') |
| 46×20-04-6 | ● | ● | ● | UF | *C. maxima* 'Hirado Buntan' × *C. reticulata* 'Cleopatra' |
| 2247×6070-02-2 | ● | ● | ● | UF | *C. reticulata* 'Nova' + *C. máxima* 'Hirado Buntan' × *C. aurantium* + *P. trifoliata* 'Flying Dragon' |
| 46×20-04-42 | ● | | | UF | *C. maxima* 'Hirado Buntan' × *C. reticulata* 'Cleopatra' |
| Orange 14 | | ● | | UF | *C. reticulata* 'Nova' + *C. maxima* 'Hirado Buntan' × *C. reticulata* 'Cleopatra' + *P. trifoliata* 'Argentine' |
| Orange 16 | ● | | | UF | *C. reticulata* 'Nova' + *C. maxima* 'Hirado Buntan' × *C. reticulata* 'Cleopatra' + *P. trifoliata* 'Argentine' |
| Sour orange | ● | | ● | NH | *C. aurantium* |
| Willits citrange | | ● | ● | USDA | *P. trifoliata* × *C. sinensis* 'Ruby' |
| Kuharske citrange | ● | ● | ● | USDA | Natural seedling variant of Carrizo |
| US-802 | ● | ● | ● | USDA | *C. grandis* Osbeck 'Siamese' × *P. trifoliata* |
| US-812 | ● | ● | ● | USDA | *C. reticulata* 'Sunki' × *P. trifoliata* 'Benecke' |
| US-897 | ● | ● | ● | USDA | *C. reticulata* 'Cleopatra' × *P. trifoliata* 'Flying Dragon' |
| US-942 | ● | ● | ● | USDA | *C. reticulata* 'Sunki' × *P. trifoliata* 'Flying Dragon' |
| Cunningham citrange | ● | ● | ● | USDA | *P. trifoliata* × *C. sinensis* |
| x-639 | ● | ● | ● | ARC-ITSC | *C. reticulata* 'Cleopatra' × *P. trifoliata* 'Rubidoux' |
| C-22 ('Bitters') | ● | ● | ● | UCR | *C. reticulata* 'Sunki' × *P. trifoliata* 'Swingle' |
| C-54 ('Carpenter') | ● | ● | ● | UCR | *C. reticulata* 'Sunki' × *P. trifoliata* 'Swingle' |

+ indicates somatic hybridization (allotetraploid). × indicates sexual hybridization (diploid or tetraploid). ARC-ITSC = Agricultural Research Council—Institute for Tropical and Subtropical Crops; NH = Natural Hybrid; UCR = University of California Riverside; UF = University of Florida; USDA = U.S. Department of Agriculture; ● = rootstock was used in the trial.

*2.2. Site Description*

Trees were planted in September 2019 at the University of Florida, Institute of Food and Agricultural Sciences (UF/IFAS) Indian River Research and Education Center in Fort Pierce, Florida in the Millennium Block experimental grove in double-row raised beds. The presence of HLB was determined to be endemic in Florida since 2013, thus, experimental trees were naturally infected in the field by the *C*Las-carrier vector. The Millennium Block grove was used in 1974 as the site of the Florida Soil-Water-Atmosphere-Plant (SWAP) project to investigate the effect of soil profile modifications and drain lines [23]. Upon the completion of the SWAP project, the area remained uncultivated until the first Millennium Block variety trials started in the early 2000s. For the current Millennium Block experiment, three independent field trials were conducted, with each in a separate location within the area. 'Ray Ruby' grapefruit (1) consisted of approximately 1.3 ha on a poorly drained Oldsmar sandy soil (Spodosols), classified as sandy, siliceous, hyperthermic Alfic Arenic Alaquods [24]. The landscape consists of natural slopes ranging from 0% to 2% and has

a non-Fe-cemented spodic horizon within 76–127 cm of the surface [25]. In T2, 1.1 ha of 'Glenn' navel orange F-56-11 trees were established in an area having mostly Ankona and Farmton soil series (spodosols), and Chobee soils (Mollisols). The latter are classified as fine-loamy, siliceous, hyperthermic Typic Argiaquolls. Chobee soils are poorly drained and have a mollic epipedon (black soil with high organic carbon). Also, they are characterized by the sandy-clay loam argillic horizon within ≈50 cm of the surface [24,25]. Lastly, nearly 1 ha of 'UF 950' (T3) mandarin trees was planted in Trial 4 on Ankona and Farmton soils from the Spodosols order.

Trees were microsprinkler-irrigated and fertilized with Wedgworth 10N-2P-7K (Wedgworth's, Belle Glade, FL, USA). This fertilizer mix is 4-month controlled-release polymer-coated sulfate of potash and polymer-coated urea at annual rates of 1.34 and 2.72 kg per tree during 2020 and 2021, respectively. Weed management and insect control were performed according to UF/IFAS Extension recommendations for citrus production [26,27]. Weather data were monitored by the Florida Automated Weather Network (FAWN) system utilizing the St. Lucie West weather station to collect total rainfall, and air temperature (Figure S1).

### 2.3. Experimental Design

Each experiment per trial was arranged in a completely randomized design consisting of linear 5-tree plots and 6 replicates per treatment (30 trees total) for the grapefruit and navel orange trials, whereas for the mandarins only 5 replications were planted because of limited material availability.

### 2.4. Data Collection and Analysis

Trunk diameter, canopy volume, disease resilience (CLas titer and HLB severity index), and leaf nutrient concentrations were assessed using the three central trees of each plot. The following variables were measured annually during the fall to determine tree size: trunk diameter at 5 cm above the bud union (perpendicular to the tree row) using an electronic digital caliper H-7352 (Uline, Pleasant Prairie, WI, USA); tree height, and canopy width along and across the row to estimate canopy volume [28]. CLas titer in leaf tissue and HLB severity index were assessed in September in 2020 and 2021. As for CLas titer, 12–18 leaves per plot were randomly collected, placed in zip-lock plastic bags, and sent to Southern Gardens Diagnostic Laboratory in Clewiston, FL for detection of HLB. Samples with a cycle threshold (Ct) value of less than 38 were considered as CLas positive. An HLB severity index rating system was created to quantify the intensity of disease symptoms based on the scale developed by Slinski [29] to score each quadrant (Figure 1). Then, an average tree disease score per canopy was produced as follows: 0 = No foliar HLB symptoms visible; 1 = Foliar disease symptoms on <20% of the quadrant. A dense quadrant with no twig dieback and minimal blotchy mottle; 2 = Foliar disease symptoms on 20 to 40% of the quadrant. Dense quadrant, some twig dieback, some blotchy mottle and possibly some tufted growth on the canopy; 3 = Foliar disease symptoms on 40 to 60% of the quadrant. Thinning quadrant with noticeable twig dieback and a few areas of the open canopy. Blotchy mottle of leaves is common, and some tufted growth is apparent; 4 = Foliar disease symptoms on 60 to 80% of the quadrant. Abundantly thin quadrant with obvious twig dieback. Most branches have blotchy mottle and/or tufted growth; 5 = Foliar disease symptoms on >80% of the quadrant. The decline has resulted in dieback of large branches and the remaining leaves are small, have blotchy mottle and may be deformed. Leaf nutrient concentrations were evaluated before the fall fertilization. After field collection, sampled leaves were rinsed with deionized water and then dried at 80 °C overnight. After drying, samples were ground using a Thomas Wiley mill (Thomas Scientific, Swedesboro, NJ, USA) and collected in a 20 mL vial. Nutrient concentrations were determined using an inductively coupled argon plasma emission (ICP-MS) spectrophotometer (Spectro Ciros CCD, Fitzburg, MA, USA). The analysis were performed at the Waters Agricultural Laboratory located in Camilla, GA (USA).

Data were analyzed annually for two consecutive growing seasons (2019–2020 and 2020–2021) using the software Rstudio version 1.3.1073 [30]. A one-way analysis of variance (function *ea2* in the easyanova package) was conducted for tree size and leaf nutrient concentration, with rootstock entered as the main effect. The residuals were checked for normality and homogeneity of variance. Square root transformations were executed as needed. *C*Las DNA concentration and HLB visual rating did not meet normality and linearity assumptions, thus, Kruskal-Wallis test was used to evaluate the main effect. When differences between treatments were significant ($p \leq 0.05$), a Tukey HSD test was used.

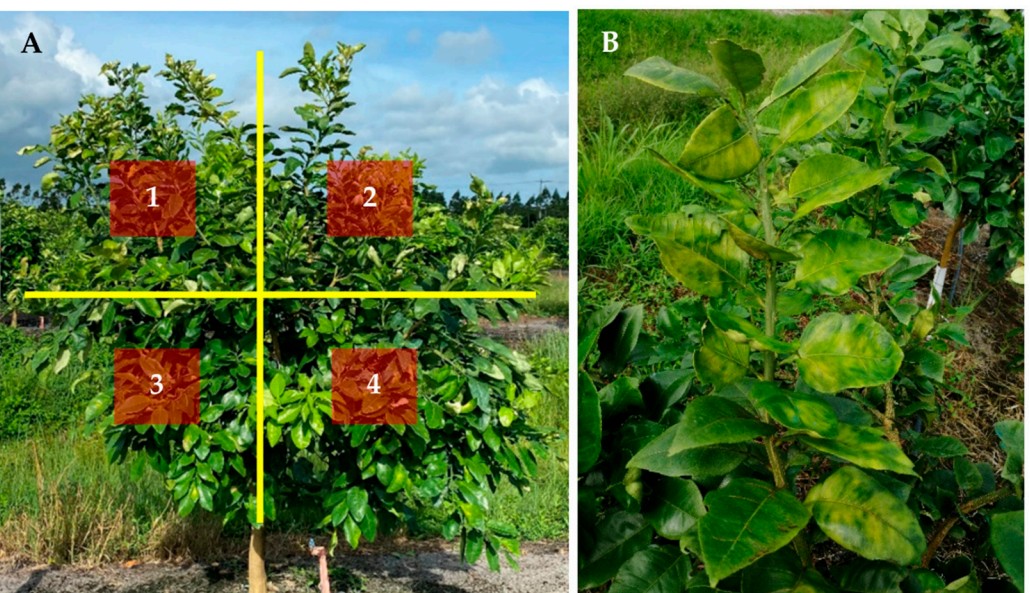

**Figure 1.** (**A**) Huanglongbing (HLB) severity rating method. Four equal quadrants were assessed for HLB severity on the east and west side of each tree. The ratings started from the top left quadrant following clockwise: 1, 2, 4, and 3. (**B**) Vegetative branch showing characteristic HLB blotchy mottling, chlorosis on leaves. (Pictures were taken at the UF/IFAS Indian River Research and Education Center, located in Fort Pierce, FL, USA).

## 3. Results and Discussions

### 3.1. Tree Size

Tree size increased over time in all treatments and was significantly influenced by the rootstock ($p < 0.001$) in both years. Trunk diameter was significantly correlated ($p < 0.001$) with canopy volume in all trials (Figure 2).

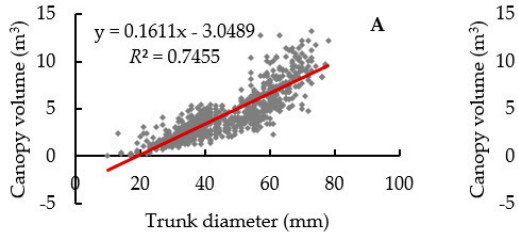 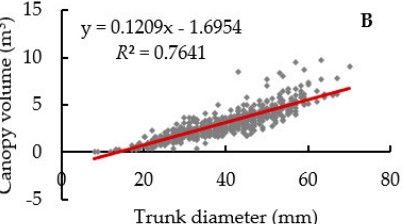 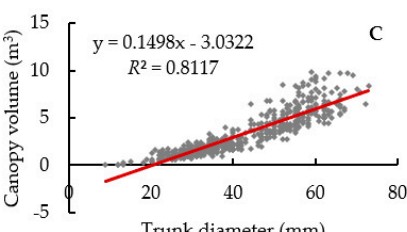

**Figure 2.** Linear regression of canopy volume (*y* axis) vs. trunk diameter (*x* axis) in 'Ray Ruby' grapefruit (**A**), 'Glenn' navel orange F−56−11 (**B**), and 'UF 950' mandarin (**C**) trial. Red line represents the trend of the linear regression.

UFR-15 induced the largest 'Ray Ruby' trees during both years of evaluation. Alternatively, Orange 16 and UFR-17 were among the rootstocks producing the smallest trees (1.2–1.4 m$^3$) in 2020, whereas 46×20-04-6 and UFR-17 developed the smallest trees (3.4–3.5 m$^3$) in 2021. Trunk diameters of 'Ray Ruby' trees were the largest (39.9–42.1 mm)

on 46×20-04-42, x-639 and A+Volk×Orange-19-11-8 in 2020. The following year, trunk diameters were the greatest (64.2–66.3 mm) in trees on UFR-15, US-812 and Cunningham citrange (Figure 3). Previous studies have shown trees on UFR-15 that grew quickly and vigorously, producing medium-large size trees [31]. Limited performance data are available for grapefruit trees on Orange 16 and 46×20-04-6. As for UFR-17, field evaluations in flatwoods conditions showed that scion cultivars grow quickly, but trees remained medium-sized, making them suitable for high-density plantings [31].

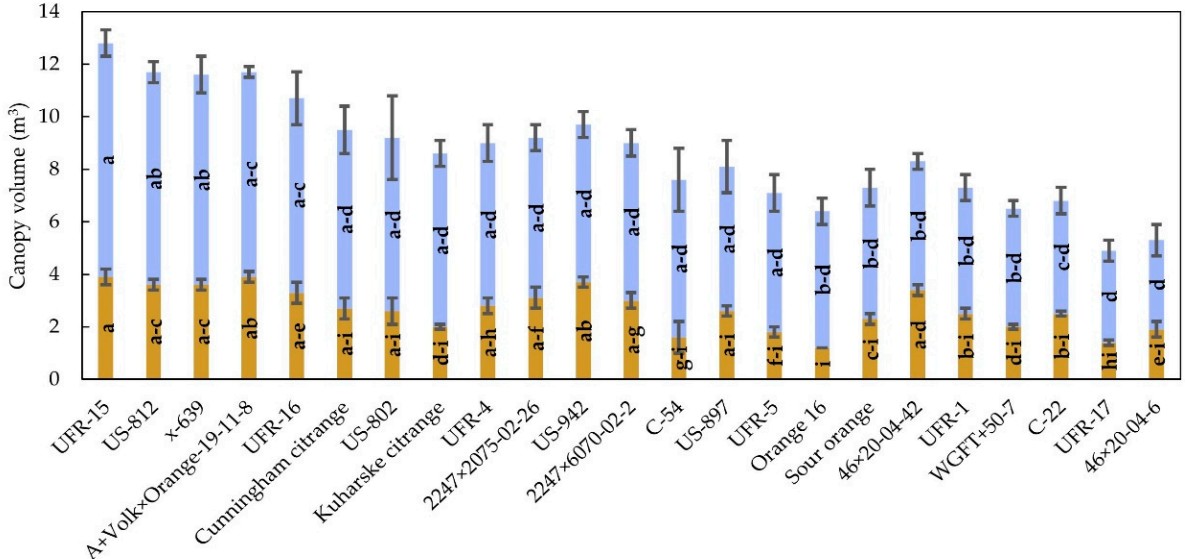

**Figure 3.** Canopy volume of 'Ray Ruby' grapefruit on each rootstock evaluated in 2020 (golden bars) and 2021 (blue bars). Mean standard errors followed by at least one common letter (centered within bars) are not significantly different by Tukey's test at $p < 0.05$. Values are back-transformed from the square root and Tukey ladder of powers.

On the other hand, the largest 'Glenn' navel orange F-56-11 trees grew on C-22 and US-802 in the years 2020 and 2021, respectively (Figure 4). Kunwar et al. [32] observed C-22 inducing medium size sweet orange trees; however, other studies have indicated that C-22 is a small size-inducing rootstock [33]. Willits citrange produced the smallest canopy volume (0.9–2.9 m$^3$) and was among the rootstocks having the thinnest trunk diameters (23.1–41.7 mm) throughout the evaluation (Figure 4). Similarly, Castle and Phillips [34] observed semi-dwarfing potential of Willits citrange when grafted with 'Valencia' sweet orange in Florida flatwoods. The largest trunk diameters were found in C-22 (34.4 mm) trees and Kuharske citrange (54.2 mm) in the years 2020 and 2021, respectively.

US-942 and US-812 produced the largest 'UF 950' mandarin trees (2.0 m$^3$) in 2020. The subsequent year, US-942 remained among the largest size-inducing rootstocks along with US-802 for 'UF 950' mandarin trees (Figure 5). Bowman et al. [35] found that trees on US-812 normally produced medium size canopies, whereas US-802 and US-942 rootstocks generated vigorous and large trees. Other studies that included US-802, US-812, and US-942 showed similar results [7,19]. WGFT+50-7 and 46×20-04-6 rootstocks generated the smallest canopy volumes (0.9–1.0 m$^3$) in 2020. Trees on UFR-1 and WGFT+50-7 had small trunk diameters (24.0–24.5 mm) in 2020, and on 46×20-04-6 rootstock in the following year (Figure 5). In another field trial, WGFT+50-7 produced small-sized trees of 'Hamlin' sweet orange [32].

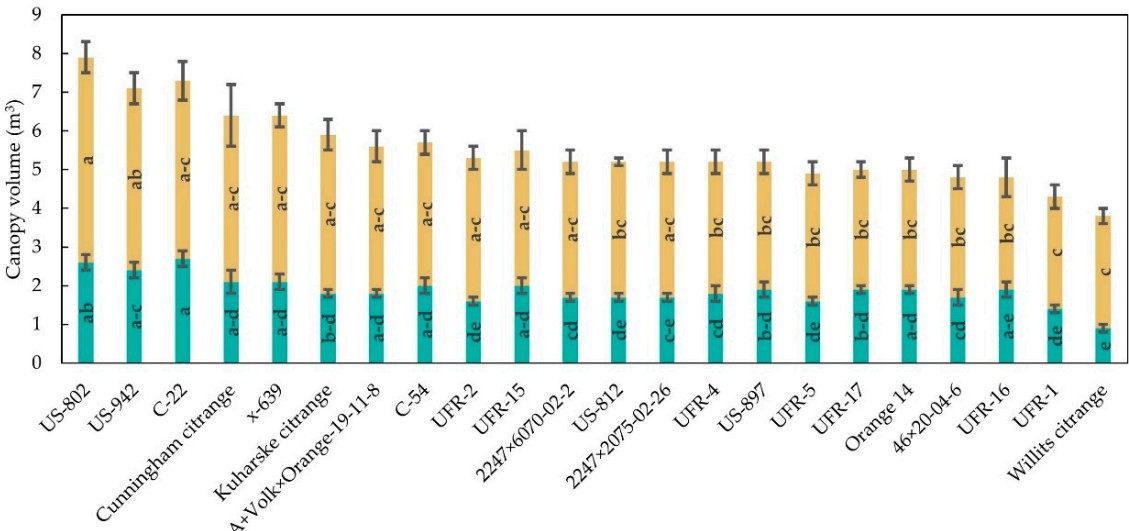

**Figure 4.** Canopy volume of 'Glenn' navel orange F-56-11 on each rootstock evaluated in 2020 (green bars) and 2021 (brown bars). Mean standard errors followed by at least one common letter (centered within bars) are not significantly different by Tukey's test at $p < 0.05$. Values are back-transformed from the square root and Tukey ladder of powers.

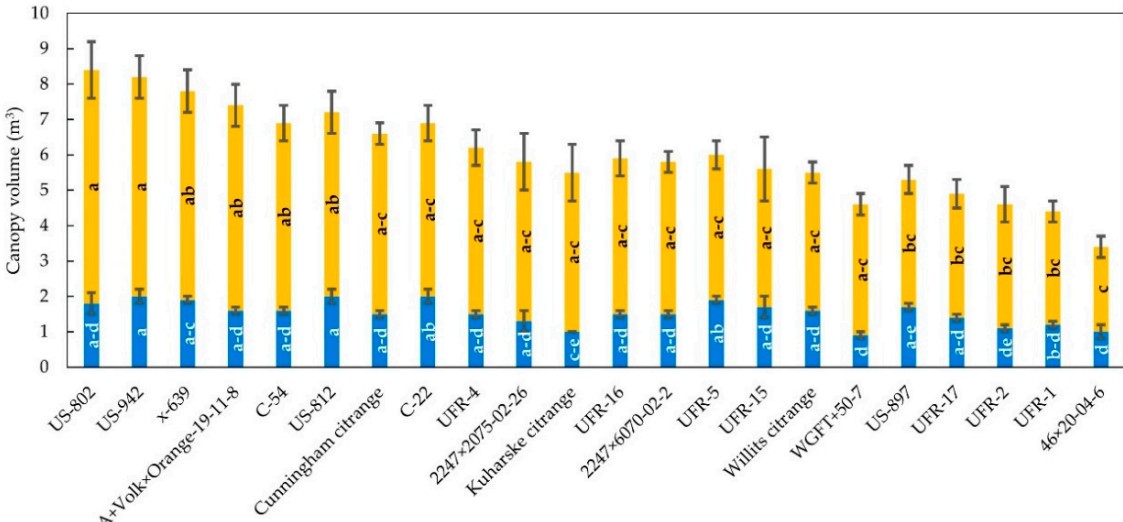

**Figure 5.** Canopy volume of 'UF 950' mandarin on each rootstock evaluated in 2020 (blue bars) and 2021 (yellow bars). Mean standard errors followed by at least one common letter (centered within bars) are not significantly different by Tukey's test at $p < 0.05$. Values are back-transformed from the square root and Tukey ladder of powers.

### 3.2. CLas Titer in Plant Leaf Tissue and HLB Severity Index

The cycle threshold of CLas DNA was not affected by the rootstock in any trial except for trial 3 (Tables 2–4). 'UF 950' mandarin trees on UFR-1 and UFR-2 produced significantly greater Ct values (39.4 and 38.3, respectively) than those growing on UFR-4 (32.4) when evaluated in September 2020 (Table 4).

Although trees in each trial showed low Ct values (indicating bacterial infection), they did not present HLB symptoms during the first year of growth. Nonetheless, nearly half of the 'UF 950'-rootstock combinations showed HLB symptoms during the rating in September 2020 through the end of the evaluation (Table 4).

During September 2021, Ct values and HLB severity index did not differ significantly among treatments in each trial (Tables 2–4). Overall, trees in each trial grew relatively well

during the first two years of evaluation despite *Clas* infection. In September 2021, HLB symptoms were observed on all 'Ray Ruby' trees evaluated with those on WGFT+50-7 rootstock showing the lowest disease index (0.4). Alternatively, trees on C-22 showed the most evident HLB symptoms (1.8) according to the scale used. WGFT+50-7 is a tetraploid somatic hybrid citrumelo that contains trifoliate orange parentage, which is widely considered highly tolerant to HLB [36].

In the navel orange trial, the disease severity index ranged from 0.2–0.5 and 0.2–1.2 during the ratings performed in March 2021 and September 2021, respectively (Table 3). Overall, the HLB severity index of 'Glenn' navel orange F-56-11 increased over time, except for those trees on Willits citrange, which HLB symptoms rated greater in Mar. 2021 than in September 2020. Willits citrange has a parentage composition (trifoliate orange) that may impart the growing scions enhanced HLB tolerance.

The mandarin trees showed the lowest HLB severity index (0.12–0.26) during the evaluation. Stover et al. [14] observed markedly greater growth and cropping of several mandarin hybrids in the IRD despite *CLas* infection. Trees on 46×20-04-6 rated the highest HLB index in March 2021 and September 2021 (0.8 and 0.7, respectively). Similarly, trees on US-942 had the most evident HLB symptoms when visually assessed in September 2020 (Table 4). US-942 is considered an HLB-tolerant rootstock. Previous studies showed sweet orange trees on US-942 with mild foliar HLB symptoms, which has not affected early tree growth after *CLas* infection [19].

Trees flushed and built vigorous canopies, especially in the 'Ray Ruby' grapefruit trial. Although experiments were independently studied, we observed that initial development of HLB in grapefruit was much slower (higher Ct values) than in other scions.

*CLas*-infected trees do not always show symptoms and become a source of inoculum simultaneously [37,38]. After infection, it may take up to 6 months or longer for a plant to show the very first symptoms (blotchy mottle), complicating the evaluation of HLB severity [4,6,38].

**Table 2.** Ct value of *CLas* DNA and HLB severity index on different 'Ray Ruby'-rootstock combinations assessed in 2020 and 2021.

| Rootstock | September 2020 | | September 2021 | |
|---|---|---|---|---|
| | Ct Value of *C*Las DNA | HLB Severity Index [y] | Ct Value of *C*Las DNA | HLB Severity Index [y] |
| 2247×2075-02-26 | 38.0 ± 1.3 [z] | 0.0 ± 0.0 | 36.7 ± 0.9 | 1.2 ± 0.3 |
| 2247×6070-02-2 | 39.3 ± 0.3 | 0.0 ± 0.0 | 38.9 ± 0.7 | 0.5 ± 0.2 |
| 46×20-04-42 | 37.0 ± 1.5 | 0.0 ± 0.0 | 38.2 ± 0.8 | 1.4 ± 0.4 |
| 46×20-04-6 | 38.3 ± 1.5 | 0.0 ± 0.0 | 39.8 ± 0.1 | 0.6 ± 0.2 |
| A+Volk×Orange-19-11-8 | 37.3 ± 1.3 | 0.0 ± 0.0 | 37.6 ± 0.9 | 1.5 ± 0.4 |
| C-22 | 36.1 ± 1.3 | 0.0 ± 0.0 | 35.9 ± 0.9 | 1.8 ± 0.4 |
| C-54 | 39.7 ± 0.3 | 0.0 ± 0.0 | 37.6 ± 0.9 | 1.0 ± 0.5 |
| Cunningham citrange | 35.4 ± 1.3 | 0.0 ± 0.0 | 39.5 ± 0.4 | 0.9 ± 0.1 |
| Kuharske citrange | 36.7 ± 0.8 | 0.0 ± 0.0 | 38.9 ± 0.5 | 0.9 ± 0.3 |
| Orange 16 | 37.1 ± 0.8 | 0.0 ± 0.0 | 36.6 ± 1.1 | 0.6 ± 0.2 |
| Sour orange | 39.1 ± 0.4 | 0.0 ± 0.0 | 36.7 ± 0.9 | 1.3 ± 0.4 |
| UFR-1 | 36.5 ± 0.7 | 0.0 ± 0.0 | 37.2 ± 1.0 | 0.8 ± 0.2 |
| UFR-15 | 35.8 ± 1.7 | 0.0 ± 0.0 | 36.8 ± 1.3 | 0.7 ± 0.2 |
| UFR-16 | 36.9 ± 2.0 | 0.0 ± 0.0 | 35.8 ± 0.9 | 1.7 ± 0.4 |
| UFR-17 | 36.9 ± 1.2 | 0.0 ± 0.0 | 37.5 ± 0.8 | 0.7 ± 0.2 |
| UFR-4 | 36.6 ± 1.9 | 0.0 ± 0.0 | 36.0 ± 1.0 | 0.8 ± 0.3 |
| UFR-5 | 35.9 ± 1.5 | 0.0 ± 0.0 | 38.2 ± 0.7 | 0.9 ± 0.2 |
| US-802 | 37.4 ± 1.5 | 0.0 ± 0.0 | 37.3 ± 1.2 | 1.1 ± 0.2 |
| US-812 | 36.8 ± 1.8 | 0.0 ± 0.0 | 38.4 ± 0.7 | 1.5 ± 0.3 |
| US-897 | 35.8 ± 1.3 | 0.0 ± 0.0 | 36.8 ± 0.9 | 1.3 ± 0.2 |
| US-942 | 37.3 ± 0.7 | 0.0 ± 0.0 | 37.7 ± 1.1 | 1.4 ± 0.4 |
| WGFT+50-7 | 38.0 ± 1.1 | 0.0 ± 0.0 | 39.7 ± 0.3 | 0.4 ± 0.2 |
| x-639 | 35.8 ± 1.5 | 0.0 ± 0.0 | 39.9 ± 0.1 | 1.4 ± 0.1 |
| *F* value | 1.11 [ns] | - | 1.96 [ns] | 1.81 [ns] |

[z] Means ± standard error (*n* = 6). [y] Foliar disease symptoms were visually rated on a scale of 0 to 5, with 0 representing a healthy canopy and 5 representing the most disease symptoms. ns = nonsignificant at $p < 0.05$. - Blank space. Ct values < 38 indicate *Clas*-infection.

**Table 3.** Ct value of *C*Las DNA and HLB severity index on different 'Glenn' navel orange F-56-11-rootstock combinations assessed in 2020 and 2021.

| Rootstock | September 2020 | | September 2021 | |
|---|---|---|---|---|
| | Ct Value of *C*Las DNA | HLB Severity Index [y] | Ct Value of *C*Las DNA | HLB Severity Index [y] |
| 2247×2075-02-26 | 36.3 ± 2.2 [z] | 0.0 ± 0.0 | 35.0 ± 1.3 | 0.9 ± 0.3 |
| Cunningham citrange | 39.1 ± 0.1 | 0.0 ± 0.0 | 36.7 ± 1.3 | 0.3 ± 0.3 |
| Orange 14 | 38.1 ± 1.0 | 0.0 ± 0.0 | 34.1 ± 0.8 | 0.8 ± 0.3 |
| UFR-1 | 38.5 ± 0.2 | 0.0 ± 0.0 | 36.1 ± 1.1 | 0.7 ± 0.2 |
| UFR-15 | 36.9 ± 1.1 | 0.0 ± 0.0 | 33.7 ± 0.8 | 1.0 ± 0.3 |
| UFR-4 | 34.8 ± 1.4 | 0.0 ± 0.0 | 37.0 ± 1.1 | 0.8 ± 0.4 |
| UFR-5 | 38.5 ± 0.1 | 0.0 ± 0.0 | 39.4 ± 0.6 | 0.8 ± 0.2 |
| US-812 | 38.9 ± 0.3 | 0.0 ± 0.0 | 35.9 ± 0.9 | 0.9 ± 0.2 |
| Willits citrange | 37.4 ± 1.6 | 0.0 ± 0.0 | 36.3 ± 1.0 | 0.2 ± 0.1 |
| x-639 | 35.3 ± 1.0 | 0.0 ± 0.0 | 34.4 ± 0.6 | 1.2 ± 0.3 |
| US-897 | 36.6 ± 2.1 | 0.0 ± 0.0 | 36.0 ± 1.1 | 1.1 ± 0.2 |
| US-942 | 38.7 ± 0.2 | 0.0 ± 0.0 | 34.6 ± 0.7 | 0.6 ± 0.2 |
| UFR-17 | 38.3 ± 1.2 | 0.0 ± 0.0 | 37.1 ± 1.2 | 0.8 ± 0.2 |
| C-54 | 39.2 ± 0.3 | 0.0 ± 0.0 | 37.7 ± 0.9 | 1.2 ± 0.2 |
| Kuharske citrange | 37.3 ± 1.4 | 0.0 ± 0.0 | 37.6 ± 0.9 | 0.8 ± 0.3 |
| 46×20-04-6 | 35.7 ± 1.7 | 0.0 ± 0.0 | 35.8 ± 1.5 | 1.1 ± 0.1 |
| A+Volk×Orange-19-11-8 | 37.4 ± 1.0 | 0.0 ± 0.0 | 33.7 ± 0.6 | 1.0 ± 0.3 |
| US-802 | 36.9 ± 1.2 | 0.0 ± 0.0 | 33.4 ± 0.8 | 0.9 ± 0.3 |
| C-22 | 35.2 ± 1.8 | 0.0 ± 0.0 | 33.2 ± 0.6 | 1.3 ± 0.3 |
| UFR-2 | 38.3 ± 0.5 | 0.0 ± 0.0 | 34.1 ± 0.8 | 1.0 ± 0.2 |
| 2247×6070-02-2 | 37.7 ± 1.3 | 0.0 ± 0.0 | 37.7 ± 1.1 | 0.5 ± 0.2 |
| UFR-16 | 36.5 ± 1.2 | 0.0 ± 0.0 | 35.1 ± 1.2 | 1.1 ± 0.4 |
| *F* value | 1.01 [ns] | - | 2.43 [ns] | 0.99 [ns] |

[z] Means ± standard error (*n* = 6) followed by at. [y] Foliar disease symptoms were visually rated on a scale of 0 to 5, with 0 representing a healthy canopy and 5 representing the most disease symptoms. ns = nonsignificant at $p < 0.05$. - Blank space. Ct values < 38 indicate *C*las-infection.

**Table 4.** Ct value of *C*Las DNA and HLB severity index on different 'UF 950'-rootstock combinations assessed in 2020 and 2021.

| Rootstock | September 2020 | | September 2021 | |
|---|---|---|---|---|
| | Ct Value of *C*Las DNA | HLB Severity Index [y] | Ct Value of *C*Las DNA | HLB Severity Index [y] |
| 2247×2075-02-26 | 36.5 ± 0.6 [z,] ab | 0.0 ± 0.0 | 34.9 ± 4.4 | 0.3 ± 0.3 |
| 2247×6070-02-2 | 35.1 ± 0.2 ab | 0.0 ± 0.0 | 34.7 ± 3.6 | 0.0 ± 0.0 |
| 46×20-04-6 | 32.8 ± 2.3 ab | 0.2 ± 0.0 | 32.8 ± 1.7 | 0.7 ± 0.1 |
| A+Volk×Orange-19-11-8 | 36.1 ± 1.5 ab | 0.0 ± 0.0 | 37.8 ± 4.4 | 0.3 ± 0.2 |
| C-22 | 33.8 ± 2.3 ab | 0.4 ± 0.0 | 38.0 ± 4.0 | 0.1 ± 0.1 |
| C-54 | 36.1 ± 1.9 ab | 0.1 ± 0.0 | 37.9 ± 3.9 | 0.3 ± 0.1 |
| Cunningham citrange | 38.1 ± 0.9 ab | 0.0 ± 0.0 | 35.3 ± 4.7 | 0.3 ± 0.3 |
| Kuharske citrange | 36.3 ± 0.5 ab | 0.3 ± 0.0 | 36.1 ± 5.6 | 0.0 ± 0.0 |
| UFR-1 | 39.4 ± 0.3 a | 0.0 ± 0.0 | 34.5 ± 4.1 | 0.0 ± 0.0 |
| UFR-15 | 34.7 ± 0.3 ab | 0.0 ± 0.0 | 40.0 ± 0.0 | 0.0 ± 0.0 |
| UFR-17 | 35.9 ± 1.1 ab | 0.1 ± 0.0 | 36.3 ± 4.0 | 0.3 ± 0.3 |
| UFR-2 | 38.3 ± 0.2 a | 0.2 ± 0.0 | 38.8 ± 2.6 | 0.4 ± 0.4 |
| US-802 | 33.8 ± 1.7 ab | 0.0 ± 0.0 | 39.3 ± 1.4 | 0.2 ± 0.2 |
| US-812 | 33.9 ± 1.7 ab | 0.1 ± 0.0 | 38.5 ± 3.3 | 0.1 ± 0.1 |
| WGFT+50-7 | 34.4 ± 1.6 ab | 0.0 ± 0.0 | 36.3 ± 4.1 | 0.0 ± 0.0 |
| x-639 | 34.6 ± 2.7 ab | 0.0 ± 0.0 | 34.3 ± 3.5 | 0.2 ± 0.1 |
| UFR-4 | 32.4 ± 1.4 b | 0.2 ± 0.0 | 34.1 ± 3.7 | 0.1 ± 0.1 |
| UFR-5 | 33.0 ± 2.2 ab | 0.2 ± 0.0 | 37.8 ± 3.4 | 0.2 ± 0.1 |
| UFR-16 | 35.0 ± 0.2 ab | 0.1 ± 0.0 | 34.6 ± 3.8 | 0.0 ± 0.0 |
| US-897 | 36.0 ± 0.7 ab | 0.7 ± 0.0 | 36.3 ± 3.4 | 0.1 ± 0.1 |
| US-942 | 35.8 ± 1.4 ab | 0.0 ± 0.0 | 38.1 ± 3.4 | 0.1 ± 0.1 |
| *F* value | 2.2 ** | 1.60 [ns] | 1.20 [ns] | 1.29 [ns] |

[z] Means ± standard error (*n* = 6) followed by at least one common letter are not significantly different at $p < 0.05$ by Tukey's honestly significant difference test. [y] Foliar disease symptoms were visually rated on a scale of 0 to 5, with 0 representing a healthy canopy and 5 representing the most disease symptoms. ** $p < 0.01$. ns = nonsignificant at $p < 0.05$. - Blank space. Ct values < 38 indicate *C*Las-infection.

*3.3. Leaf Nutrient Concentrations*

Leaf nutrient concentration was evaluated only in the second year of growth and varied with the rootstock used in each trial (Tables S1–S3). The scion and rootstock influenced leaf nutrient concentrations. Macro- and micronutrients, except calcium, were in the optimal range in all trials [39]. This study was located within an area having mostly Spodosols, which contain a characteristic acidic subsurface layer that might have risen to the top surface during the bedding process before tree planting. This, along with the fertilizer mix applied, might cause soil pH to drop in some areas across the field with values as low as 5.0, and as a result, calcium became less available for uptake [40]. Thus far, calcium deficiency symptoms have not been present at the canopy level, though potential effects on fruits remain to be seen.

In the 'Ray Ruby' trial, although all rootstocks induced foliar nitrogen levels within the recommended range, 2247×6070-02-2, A+Volk×Orange-19-11-8, US-942, and WGFT+50-7 inducing significantly greater N levels (30–31 g kg$^{-1}$; $p < 0.002$) than trees on UFR-16 (25.7 g kg$^{-1}$). These results are preliminary, and they can vary as trees grow. Because trees are developing in an HLB-endemic environment, thorough attention is needed to minimize the detrimental effects that can potentially be caused by the disease. On the other hand, leaf nitrogen levels were not correlated with tree size, and similar studies support the latter statement [41]. However, fertilization programs with increased nitrogen may promote excessive vegetative growth, which can make ACP populations peak and enhance CLas transmission [40,42].

## 4. Conclusions

In this study, we presented early-perfomance data related to growth under Huanglongbing (HLB) conditions of 2-year-old 'Ray Ruby' grapefruit, 'Glenn' navel orange F-56-11, and 'UF 950' mandarin trees grafted on UFR, commercial, and other recently released rootstocks. Rootstock performance differed among scions. Tree size, *CLas* titer, and leaf nutrient concentration were the variables most affected by the rootstock in each trial. Tree size varied depending on the rootstock with each citrus scion. UFR-15 produced the largest 'Ray Ruby' grapefruit trees, whereas 'Glenn' navel orange F-56-11 and 'UF 950' mandarins were larger on C-22 and US-802, respectively. Despite that most trees across trials becoming *CLas*-infected, minimum HLB visual symptoms were observed, and trees grew vigorously and remained relatively healthy over the experiment. WGFT+50-7 and Willits induced the lowest HLB symptoms in 'Ray Ruby' and 'Glenn' trees, respectively. 'UF 950' on Cunningham, Kuharske, UFR-1, UFR-15, UFR-16, WGFT+50-7, and Willits showed the lowest HLB symptoms by the end of the evaluation.

Most trees had leaf macro- and micronutrients within optimum levels, except for Ca. Although significant different N concentrations in leaves were observed in the 'Ray Ruby' trial, no correlation was detected either with tree size or *CLas* titer/HLB severity index. As trees reach maturity, tree size differences might have further implications regarding cultural practices (harvest, pruning, trimming, chemical spraying, etc.), thus impacting production costs. Longer-term studies are still required to evaluate tree performance regarding productivity, fruit quality, and HLB resilience to determine the most consistent and economically viable rootstocks for fresh-market grapefruit, navel orange and mandarin production in Florida.

**Supplementary Materials:** The following supporting information can be downloaded at: https://www.mdpi.com/article/10.3390/horticulturae8111027/s1, Figure S1: Temperature and precipitation data collected in 2019/20 (A) and 2020/21 (B) by the Florida Automated Weather Network (FAWN) using the St. Lucie West (Florida, USA) weather station. The red vertical line represents the end of the solar year; Table S1: Macro- and micronutrients concentration in the leaves of 'Ray Ruby' grapefruit on multiple rootstocks; Table S2: Macro- and micronutrients concentration in the leaves of 'Glenn' navel orange F-56-11 on multiple rootstocks; Table S3: Macro- and micronutrients concentration in the leaves of 'UF 950' mandarin on multiple rootstocks.

**Author Contributions:** Conceptualization, R.S.F., P.D.S. and W.S.C.; investigation, R.S.F., P.D.S. and J.M.Z.-M.; resources, R.S.F. and L.R.; data collection and analysis, J.M.Z.-M.; writing—original draft preparation, J.M.Z.-M.; writing—review and editing, J.M.Z.-M., L.R., R.S.F., P.D.S., W.S.C., J.W.G. and F.G.G.J.; supervision, R.S.F. and L.R.; funding acquisition, R.S.F. All authors have read and agreed to the published version of the manuscript.

**Funding:** Funding for this research was provided by the UF/IFAS Dean for Research for the purchase of citrus trees, the UF/IFAS Indian River Research and Education for operational support, and the Citrus Research and Development Foundation (CRDF, project #18-037C "Performance of new grapefruit cultivars and rootstocks in the Indian River District").

**Data Availability Statement:** Raw data are available by request via email to l.rossi@ufl.edu.

**Acknowledgments:** We thank Ed Stover, Brian Scully, the Indian River Citrus League (Pat Schirard, Doug Bournique, and Karen Smith) and numerous perseverant grapefruit growers for the initial project discussion and the list of materials to trial, Brite Leaf Citrus Nursery (Anna and Nate Jameson) for the meticulous work with the plant material used in the trials, Mac Hossain, Christopher Hernandez, Cristina Gil, Liz Calise, Beth Curry, Tom James, Laura Cano, Eduardo Esteves, Flavia Zambon, Randy Burton and Steve Mayo for their help in data collection, data analysis, and management, CRDF and the Board members for funding the project, Tom Stopyra for the technical support, and Ronald D. Cave (grant co-PI) and the UF/IFAS Indian River Research and Education Center personnel for the field and administrative assistance with the Millennium Block grove.

**Conflicts of Interest:** The authors declare no conflict of interest. The funding agencies had no role in the design of the study, in the collection, analyses, or interpretation of data, in the writing of the manuscript, or in the decision to publish the results.

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
