# Peer review of "Early Performance of Recently Released Rootstocks with Grapefruit, Navel Orange, and Mandarin Scions under Endemic Huanglongbing Conditions in Florida"

_horticulturae, doi:10.3390/horticulturae8111027_

Round 1

Reviewer 1 Report

The manuscript title “Early Performance of Recently Released Rootstocks with Grapefruit, Navel Orange, and Mandarin Scions for Florida Fresh Citrus” is a good initial study to find-out a good rootstock/scion combination against HLB disease. This MS need minor improvements of method section.

Comments for author:

1-      How the trees, were inoculated with HLB bacteria? Please explain this clearly in the methods section. The scion was inoculated with HLB bacteria? Or the rootstock? Is it optimized?

2-      I suggest authors to mention HLB in the title, because authors check the early performance against HLB.

3-      The author frequently discussed about the “Foliar disease symptoms” in the abstract, results, discussion, and conclusion. However, they didn’t add a single picture of foliar disease symptoms? Why? If the authors add the symptoms figure, than it will increase the impact of the paper.

4-      The weather data should be given as a supplementary file.

Author Response

The manuscript title “Early Performance of Recently Released Rootstocks with Grapefruit, Navel Orange, and Mandarin Scions for Florida Fresh Citrus” is a good initial study to find-out a good rootstock/scion combination against HLB disease. This MS need minor improvements of method section.

Thank for your kind revisions. Below you can find are replies to your comments:

Comments for author:

1. How the trees, were inoculated with HLB bacteria? Please explain this clearly in the methods section. The scion was inoculated with HLB bacteria? Or the rootstock? Is it optimized?

HLB is considered endemic across the state of Florida, so the trees were naturally inoculated in the field. The experimental area is surrounded by citrus groves, all showing high incidence of the disease. To address this comment, more information has been added to the “site description” section.

2. I suggest authors to mention HLB in the title, because authors check the early performance against HLB.

The title has been changed and the word HLB is now included.

3. The author frequently discussed about the “Foliar disease symptoms” in the abstract, results, discussion, and conclusion. However, they didn’t add a single picture of foliar disease symptoms? Why? If the authors add the symptoms figure, than it will increase the impact of the paper.

A figure has been added.

4. The weather data should be given as a supplementary file.

The weather data has been moved to the Supplementary File.

Reviewer 2 Report

I indicate some comments that you should attend to improve the manuscript

Lines 147-148: The authors should describe the methodology and equipment used to measure the trunk diameter and leaf nutrient concentration.

Lines 193-195: This information should be deleted “However, field observations indicate that they usually are small to medium size-inducing rootstocks (J.W. Grosser, personal communication)”

Leaf nutrient concentrations section is incomplete. The results are not included in the submitted manuscript, they must be presented in a table. In addition, the authors should review the conclusions provided on these analyzes and indicate that they are preliminary results (one year).

Author Response

I indicate some comments that you should attend to improve the manuscript

Thank you for taking the time to review our manuscript, please find our replies to your comments below.

Lines 147-148: The authors should describe the methodology and equipment used to measure the trunk diameter and leaf nutrient concentration.

The methods are now described.

Lines 193-195: This information should be deleted “However, field observations indicate that they usually are small to medium size-inducing rootstocks (J.W. Grosser, personal communication)”

Deleted.

Leaf nutrient concentrations section is incomplete. The results are not included in the submitted manuscript, they must be presented in a table. In addition, the authors should review the conclusions provided on these analyzes and indicate that they are preliminary results (one year).

Conclusions were revised and corrected as recommended. Nutrient data are now included in a supplementary table.